# Motorized Intramedullary Nail Lengthening in the Older Population

**DOI:** 10.3390/jcm11175242

**Published:** 2022-09-05

**Authors:** Kenneth P. Powell, Ahmed I. Hammouda, Larysa P. Hlukha, Jessica C. Rivera, Minoo Patel, S. Robert Rozbruch, Janet D. Conway, John E. Herzenberg

**Affiliations:** 1Shriners Hospital for Children, Shreverport, LA 71103, USA; 2Department of Orthopedics and Trauma, Al-Azhar University Hospitals, Cairo 11511, Egypt; 3International Center for Limb Lengthening, Rubin Institute for Advanced Orthopedics, Sinai Hospital of Baltimore, Baltimore, MD 21215, USA; 4Centre for Limb Lengthening and Reconstruction, Richmond, VIC 3121, Australia; 5Hospital for Special Surgery, New York, NY 10021, USA

**Keywords:** bone lengthening, geriatric, intramedullary lengthening nail, limb length discrepancy, Precice

## Abstract

Limb lengthening has not been widely employed in the elderly population due to concerns that outcomes will be inferior. The purpose of this multicenter, retrospective case-control series was to report the bone healing outcomes and complications of lower limb lengthening in older patients (≥60 years) using magnetic intramedullary lengthening nail (MILN). Our hypothesis was that healing parameters including consolidation days, the consolidation index, maturation days, and the maturation index, as well as the number of adverse events reported in the older population, would be no different to those of the general adult population. We retrospectively reviewed charts and radiographs from patients ≥60 years of age with limb-length discrepancies who underwent femoral or tibial lengthening using a MILN. Parameters were compared among the age categories “≤19 years,” “20–39 years,” “40–59 years,” or “≥60 years” and propensity-matched cohorts for the age groups 20–59 years and ≥60 years. Complications were reported as percentages for each age category. In the study period, 354 MILN were placed in 257 patients. Sixteen nails were placed in patients 60 years of age or older (mean 65 ± 5 years; range 60–72 years). Comparisons of healing parameters showed no difference between those aged 60+ and the younger cohort. Complication percentages were not statistically significant (*p* = 0.816). Limb lengthening with MILN may therefore be considered a safe and feasible option for a generally healthy elderly population.

## 1. Introduction

Limb length discrepancy (LLD) can be caused by many acquired or congenital etiologies which can be corrected by distraction osteogenesis [1]. Traditionally, lengthening has not been performed in the geriatric population, defined in this case series as age > 60 years. Limb lengthening has historically been performed by external fixators which are bulky and uncomfortable, and associated with pin tract infections, joint stiffness, muscle contractures, frame failure, and soft tissue injury [2,3]. Dual hybrid system fixation methods, such as lengthening over a nail (LON) or lengthening and then nailing (LATN), were created to diminish the complications associated with conventional techniques and decrease the time spent in the external fixator frame [4,5,6]. However, these modifications do not completely eliminate the aforementioned complications [4,5,6,7].

Totally implantable intramedullary (IM) lengthening devices have been introduced as a novel option for bone lengthening. There are several types of IM lengthening devices that utilize different lengthening methods, such as a ratchet assembly (Albizzia nail), a roller-clutch threaded rod mechanism (intramedullary skeletal kinetic distractor (Orthofix, Lewisville, TX, USA)) driven by rotational movements through the osteotomy, or an electrically controlled linear actuator (Fitbone, Orthofix) [8]. The Precice magnetic intramedullary lengthening nail (MILN), introduced in 2011, uses an external remote controller unit composed of revolving magnets that cause an internal magnet inside the nail to rotate, driving a rod [9]. The Precice and other MILNs have been investigated in both adult and pediatric populations.

Several studies have assessed the use of external fixation and IM nailing to stabilize femoral fractures in the elderly population [10,11]. Shulman et al. [11] performed a retrospective review comparing the outcomes for two cohorts who underwent either IM-nail- or locking-plate-fixation of distal femoral fractures; one cohort had a mean age of 78 years and the other a mean age of 47 years. They reported no statistically significant differences in fracture union, range of motion (ROM), emotional indices, or mobility indices between the cohorts, although daily activity, functional indices, and bothersome indices were worse in the older cohort [11].

To our knowledge, there is no literature describing limb lengthening specifically in geriatric patients. The purpose of this report is to determine the outcomes of patients with MILNs aged ≥ 60 years and describe the effects of age on distraction and healing parameters. Outcomes are compared to a younger population who have also undergone lower extremity lengthening with MILNs. Our hypothesis was that bone healing parameters and the number of adverse events reported in the older population would be similar to those of the general adult population.

## 2. Materials and Methods

This review was approved by our hospital’s institutional review boards. Case logs of six orthopedic surgeons for this multicenter review were used to identify patients who underwent osteoplasty and the placement of MILNs between 2012 and 2020. Electronic medical records were queried for demographic and case/treatment data. Radiographs were routinely obtained following surgery. A radiographic review of digital films in conjunction with clinical notes determined preoperative LLD, the amount of lengthening achieved, and the date of full distraction and consolidation. Radiographs were reviewed and measurements obtained by a board-certified orthopedic surgeon other than the treating surgeon for the studied variables.

The primary predictor variable was the age of the patient at the time of surgery. Age was analyzed categorically according to four groupings: ≤19 years, 20–39 years, 40–59 years, and ≥60 years. Control variables included sex, bone lengthened (femur or tibia), nail diameter, actual lengthening achieved (cm), and the distraction index (DI). The DI was equal to the amount of lengthening (regenerate bone) (cm) divided by number of days of lengthening prescribed. The DI, therefore, yielded lengthening per day as a proxy for the prescribed distraction rate.

Healing parameter outcomes included consolidation days, the consolidation index, maturation days, and the maturation index. Consolidation days were numbered between the date of surgery and the date determined by the treating physician as suitable for full weight bearing. The consolidation index (CI) was calculated by dividing the number of consolidation days by the lengthening achieved (cm) and indicated the total treatment time per cm of length achieved. Maturation days were numbered between the date distraction was completed and the last consolidation day. The maturation index (MI) was calculated by dividing the number of maturation days by the lengthening achieved (cm) and indicated the time required to heal the regenerate bone per cm of length achieved.

Complications related to surgery, distraction, and healing were enumerated from the date of surgery to the last follow-up available in the medical record, and were analyzed as count data. Each subject was designated as having a problem, obstacle, minor complication, or major complication during the treatment course, as per Paley et al. [12].

### 2.1. Analysis

Means and confidence intervals were calculated for consolidation days, CI, maturation days, and MI. Due to the small sample size of patients aged ≥ 60 years and the unequal variances determined by Bartlett’s testing, Kruskal–Wallis tests and the post hoc Dunn’s test with Bonferroni correction for multiple comparisons were employed for univariate comparison of these healing parameters between the age categories for all subjects.

The youngest age category (≤19 years) and subjects with nail sizes or material construction that routinely influences weight-bearing restrictions were excluded from the secondary analysis. The nails excluded were small diameter nails and advanced alloy nails that may delay or accelerate, respectively, weight-bearing recommendations. Stainless steel Precice magnetic nails were withdrawn from the market due to significant complications and are not included in this study.

A comparable control group of the remaining subjects aged 20–59 years were then identified using propensity score matching. Variables such as lengthening achieved, sex, the bone lengthened, and nail size were matched to test the effect of age between the ≥60 category and the younger control group. Analysis was performed per bone lengthened rather than per patient in cases where bilateral surgery was performed. Analysis was performed using StataCorp, Stata Statistical Software, IC 14.2 (College Station, TX, USA).

### 2.2. Surgical Technique

All procedures were performed at a single institution by the attending surgeons using a standard technique, postoperative care plan, physical therapy, and follow-up schedule, as reported by Rozbruch et al. [13]. A 1 cm incision was made with soft tissue dissection at the level of the designated osteotomy site. The multiple drill hole technique was then used at the osteotomy site to vent the IM canal, to prevent fat embolism. The IM canal was then gradually reamed by 0.5 mm increments until the reamer was 2 mm over the diameter of the chosen MILN. An osteotome was used to cut at the osteotomy level and the selected MILN was guided down the canal and locked proximally and distally with the use of locking screws. The MILN was tested intraoperatively by acutely distracting 1–2 mm.

Patients were instructed to begin distraction at a rate of 1 mm/day (femur) or 0.75 mm/day (tibia) and to attend follow-up appointments at the clinic biweekly during the distraction phase and monthly during the consolidation phase. Anteroposterior (AP) and lateral radiographs of the femur/tibia were obtained at each visit, as well as long AP films to assess LLD when necessary. The rate of distraction was adjusted by the attending surgeon during the follow-up appointments as needed. Patients were instructed to perform physical therapy 3–5 times a week for 1 hour per day. The patient was to remain partially weight-bearing (30–50 lbs) using a crutch until the bone was deemed consolidated (defined as three of four cortices being determined to have healed on radiographic inspection).

## 3. Results

In the study period, 354 MILNs were placed in 259 patients. In total, 186 (53%) were placed in male patients. Sixteen MILNs (ten femora, six tibiae) were placed in patients aged ≥ 60 (mean 65 ± 5; range 60–72) years. The mean duration of overall follow-up after the index surgery was 25 (range, 1–78) months.

Six of 16 patients in the aged ≥ 60 years cohort had multiple comorbidities, such as obesity, peripheral neuropathy, coronary artery disease, renal artery stenosis with stent placement, prostate cancer, pulmonary hypertension, spinal stenosis, hypertension, depression, reflux, or a history of osteomyelitis. Patients had LLD secondary to traumatic injury, acquired or congenital deformities, or after knee fusion. Table 1 presents the characteristics of the ≥60 year cohort. In the younger group of patients aged < 60, etiologies included traumatic injury, infected nonunion, knee fusion, congenital conditions, and cosmetic concerns. The method of nail insertion varied depending on previous operations or existing hardware.

One patient in the ≥60 year cohort died during the course of treatment due to an unrelated illness. Because this subject did not progress to the phase where the lengthened bone was consolidated/matured, they are excluded from the CD, CI, MD, and MI analyses. Healing parameter means and 95% confidence intervals are shown in Table 2. The mean DI in the older population was 0.65 mm/day, while the younger population achieved a mean DI of 0.67 mm/day. The mean CI was 34 (range, 28–41) days/cm, and the mean MI was 17 (range, 12–22) days/cm. There were no differences among the age categories for the healing parameters.

Secondary analysis excluded 261 subjects in the ≤19 year age category and 6 additional subjects in the 20–39 year age category who received small diameter or stronger alloy nails. Comparisons of the healing parameters showed no differences between the ≥60 year cohort and the younger population.

Table 3 shows the ≥60-year age group’s lengthening outcomes, including post-explantation complications. Complications for the entire cohort of MILNs are reported in Table 4 with the totals and percentages of segments that experienced a complication, with some having multiple complications. In the older population, 56% of segments had a complication, with 33% experiencing more than one. In total, 53.3% of the younger population (age 7–19) reported complications, with 30.2% reflecting multiple events per segment/patient. These results did not reach statistical significance: *p* = 0.816. The 20–39- and 40–59-year-old cohorts were determined to have rates of complications of 52.7% and 59.1%, respectively (*p* = 0.804; *p* = 0.861).

### Sample Case

A 71-year-old male, an avid skier and cyclist with a history of sports-related injuries, was referred for LLD. He suffered a right femur fracture at age 53 that required femoral IM nail fixation. He also experienced a right intertrochanteric femur fracture at age 67, for which he underwent sliding screw fixation. He walked with a half-inch heel lift inside his right shoe. He had been experiencing lower back and right knee pain for more than three years prior to initial presentation. He stated that he did not want to wear shoe lifts for the rest of his life and found it difficult to maintain an active lifestyle. The patient also had a history of prostate cancer and coronary artery bypass surgery, both of which had been treated successfully. He no longer experienced cardiac symptoms. Upon physical examination, he demonstrated right femoral retroversion with a slight pelvic tilt with compensation noted. There was a femoral discrepancy (right side shorter than left) of approximately 2 cm when performing the Galeazzi test. Radiographs showed mild osteopenia with plate and screw fixation of the proximal right femur in the area of the intertrochanteric fracture, with mature bone callus formation. A 3.0 cm femoral discrepancy was measured, with the right side shorter than left side (Figure 1).

A right femoral osteoplasty was performed and a Precice MILN was inserted using the methods described above (Figure 2). The patient began lengthening as instructed on postoperative day five. He attended physical therapy and took vitamin D and calcium supplementation, as well as oxycodone as needed for pain management. Initial ROM of the knee joint was restricted from 0–50°, but full range was regained two months after surgery. He finished distraction 46 days after surgery (Figure 3). The patient completed consolidation 152 days after surgery and experienced no complications. At his latest follow-up appointment, his long leg films (Figure 4) demonstrated complete healing, remodeling, and equal limb length. The patient has returned to his active lifestyle, including skiing, and reported that he felt better on his bike and skis.

## 4. Discussion

Limb lengthening has evolved from the use of traditional systems, such as circular and unilateral external fixation, to LON or LATN, to lengthening using a completely implantable motorized IM nail. A more recent advancement involves a motorized MILN with an external remote controller for distraction [9,12]. Several studies have assessed the use of this kind of nail in the adult and pediatric populations, and have shown that this device is an accurate and viable alternative for lengthening in these populations [8,9,13]. To our knowledge, there have been no studies dedicated to reporting the use of motorized MILNs in the geriatric population (aged ≥ 60 years).

It is possible that lengthening in this older population has not been considered previously because this demographic has higher rates of comorbidities, reduced bone quality, and increased mortality rates following surgical procedures [10,11,14]. Patients who undergo surgeries at such an age (such as hip replacements and fracture stabilization) tend to be more medically complex. In patients who are relatively healthy, limb lengthening may be a viable option to promote a better quality of life.

Paley et al. [13] reported on 41 patients (54 nails), 6 of whom underwent post-traumatic limb lengthening using the Precice MILN; they had a mean age of 49 years (range, 30–58) and a preoperative lengthening goal of 3.48 cm. Overall, patients had a mean lengthening rate of 0.93 mm/day, a mean length achieved of 4.41 cm, and a mean healing time of 125.3 days. Shabtai et al. [15] reported on 18 patients (21 bone segments) with congenital limb deficiency, who demonstrated an overall healing index (CI) of 0.91 months/cm; among these patients, there were seven complications, and the study demonstrated the accuracy of the external remote control and the maintenance of the adjacent joint’s range of motion. Several other studies determined the CI for femur or tibial lengthening to be 51.4 days/cm with use of the Ilizarov and monolateral external fixators [16], 66.6 days/cm with use of the Taylor Spatial Frame (Smith & Nephew, London, UK) [17], and 43.7 days/cm with use of the Association for Osteosynthesis/Association for the Study of Internal Fixation’s external fixator (Synthes, Paoli, PA, USA) [18].

Our lengthening rate is slightly slower than the generally recommended amount of 1.0 mm/day; however, our DI and our healing indices appear to be similar to those reported in the literature [19]. This may indicate the parameters assessed are not different between the geriatric and the younger population.

Fischgrund et al. [20] reported on the results of bone-segment lengthening using Ilizarov external fixation. The effect of age on regenerate healing was noted, as patients undergoing metaphyseal tibial lengthening demonstrated significantly different distraction consolidation indices according to age group. The group aged ≥ 30 years healed slower than the group aged 20–29, while the 20–29 group healed slower than the group aged ≤ 19. The authors were unable to break their femoral segment group into age ranges for comparison. Looking at femoral and tibial segments combined, our subsets of the younger population, aged 20–39 and 40–59, showed a trend towards the equivalency of these parameters with the ≥60 population (Table 4).

Our study is not without its limitations. We had a small, non-randomized sample group which included 11 femora and five tibiae with minimal deformity, which required only bone lengthening. The stainless-steel magnetic lengthening nail was taken off the market in 2021 and, therefore, is not discussed in this manuscript. This can partially explain the small sample size, since only titanium alloy MILNs were included. Further studies will need to be performed to assess larger cohorts. Another limitation is the average length of the follow-up period of 25 (range, 1–78) months. Additional research to assess the long term follow-up of these patients is required to better understand outcomes after nail removal.

To our knowledge, this multicenter retrospective case-control series is the first to report on the outcomes of limb lengthening using MILNs in an older population as compared with a younger patient population cohort. Based on our results, the healing parameters and the number of adverse events reported in the older population are similar to those of the younger population. Further studies comparing the use of different devices in this population will aid in determining the best option for the treatment of LLD. It appears that limb lengthening with MILNs may therefore be a safe and feasible option for the generally healthy members of the geriatric population, allowing for improved quality of life. Therefore, age alone should not be a contraindication to lower limb lengthening.

## Figures and Tables

**Figure 1 jcm-11-05242-f001:**
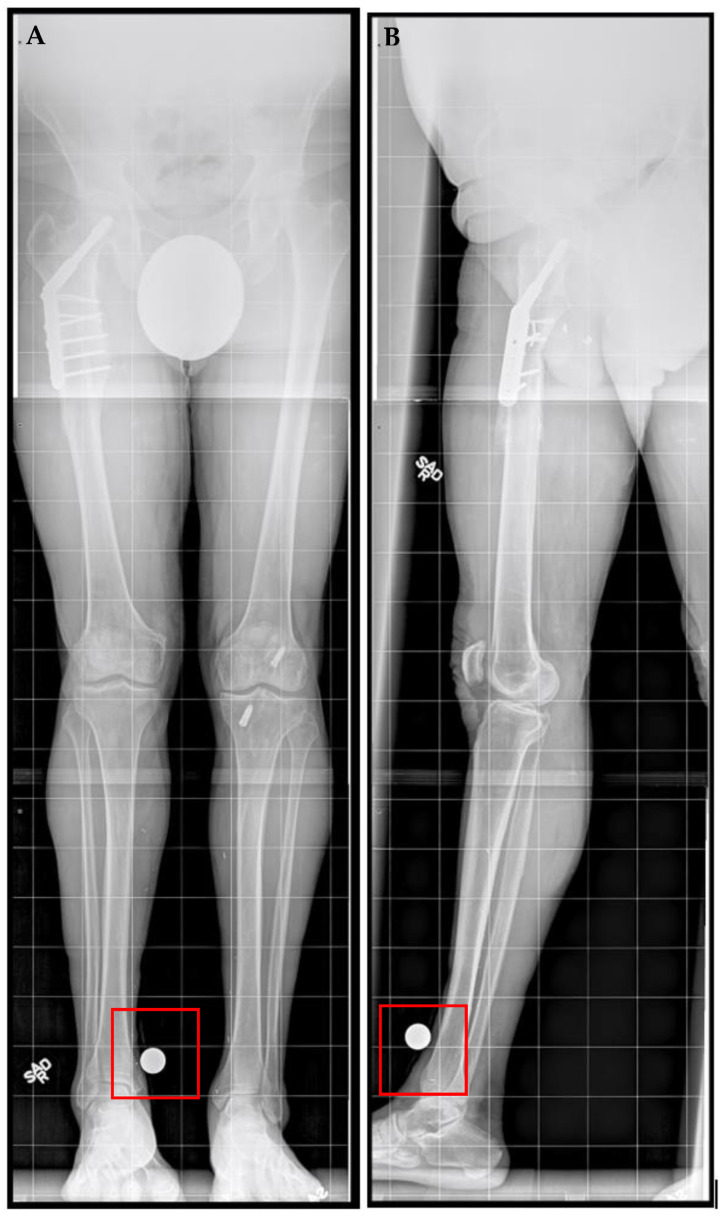
Posttraumatic femoral lengthening in a 71-year-old male patient, shown preoperatively. (**A**) Anteroposterior bilateral long leg and (**B**) right long leg lateral prior to the insertion of the Precice MILN. Gray dots on the bottom represent X-ray calibration spheres.

**Figure 2 jcm-11-05242-f002:**
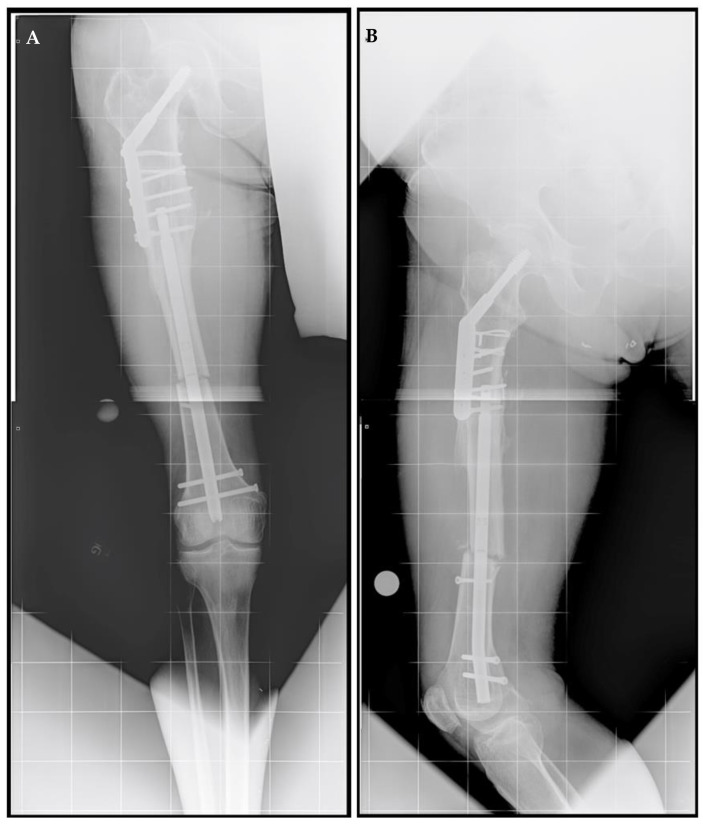
Post-traumatic femoral lengthening in a 71-year-old male patient, shown here postoperatively. (**A**) Anteroposterior and (**B**) lateral of the right femur after osteoplasty and the insertion of the Precice MILN. Gray dots on the bottom represent X-ray calibration spheres.

**Figure 3 jcm-11-05242-f003:**
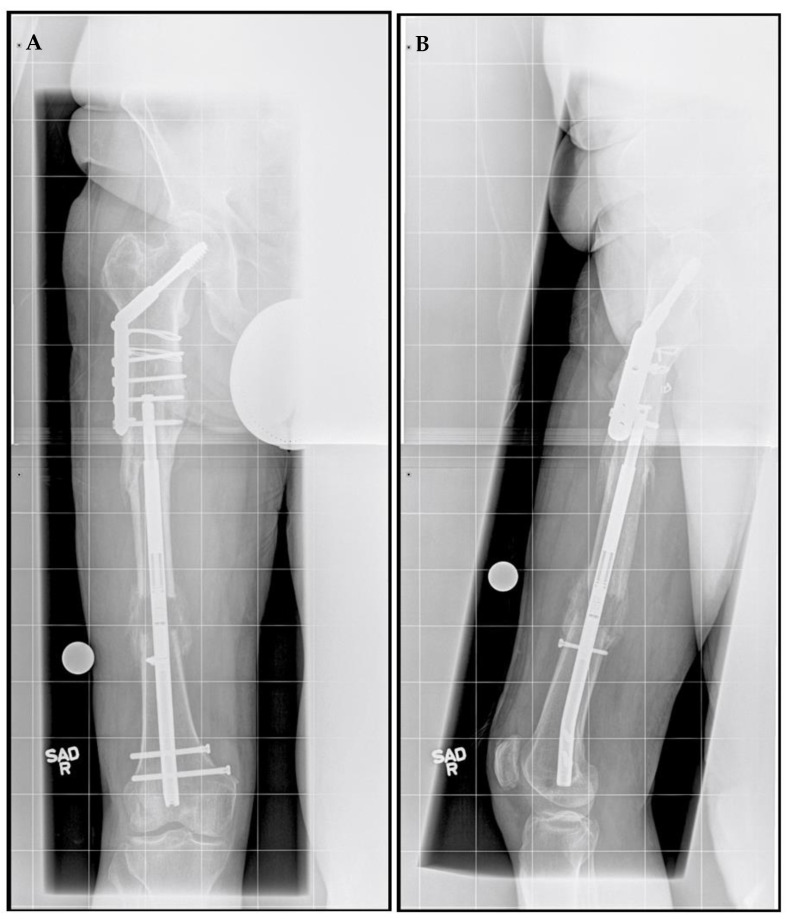
Post-traumatic femoral lengthening in a 71-year-old male patient after the distraction phase. (**A**) Antero-posterior and (**B**) lateral of the right femur after the completion of 3 cm distraction with Precice MILN. Gray dots on the bottom represent X-ray calibration spheres.

**Figure 4 jcm-11-05242-f004:**
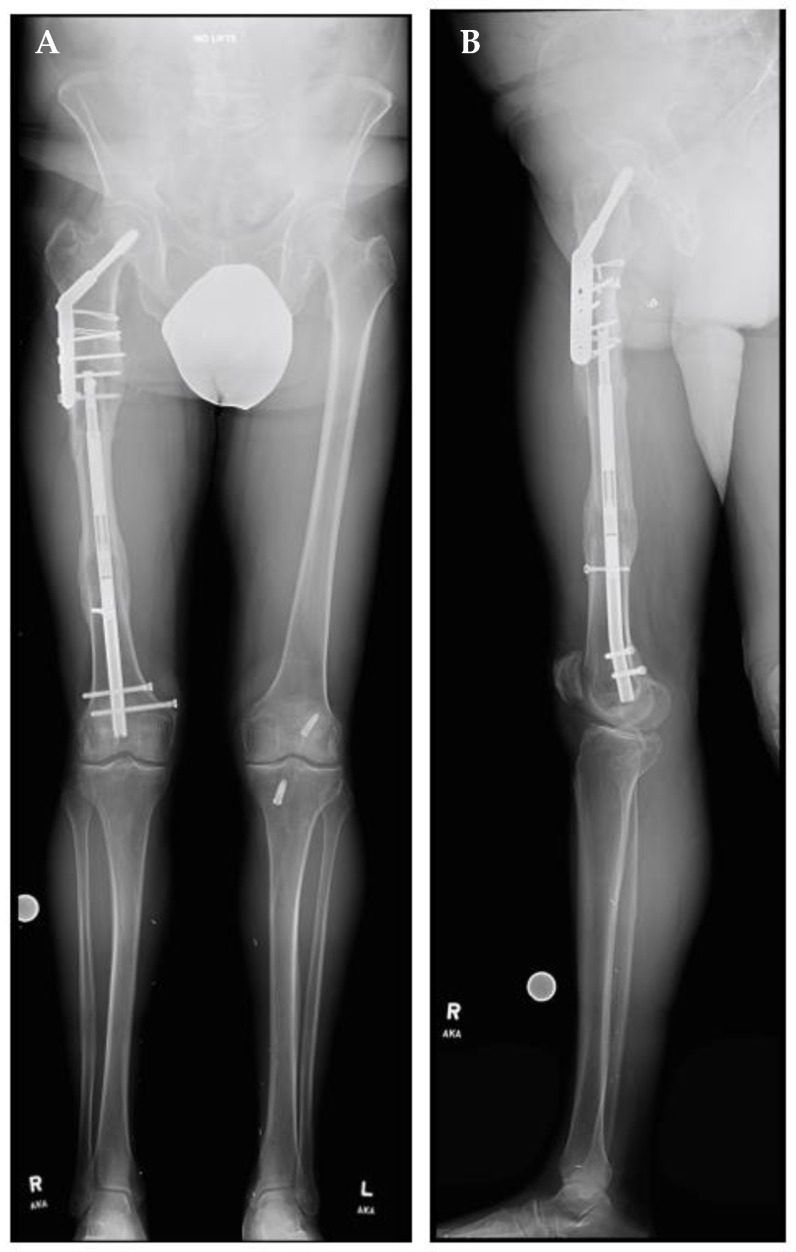
Post-traumatic femoral lengthening in a 71-year-old male patient at his final follow-up appointment. (**A**) Anteroposterior bilateral long leg and (**B**) lateral right long leg after consolidation of regenerate, with Precice MILN in place. Gray dots on the bottom represent X-ray calibration spheres.

**Table 1 jcm-11-05242-t001:** Characteristics of patients aged ≥ 60 years.

Patient	Sex	Age	Etiology of LLD	Comorbidities	Goal Lengthening (cm)	Bone Operated	Nail Insertion Method
1	M	60	Congenital		6.0	Tibia	Antegrade
2	F	60	Infected nonunion	Prior infection	4.9	Femur	Retrograde
3	M	60	Acquired		5.6	Femur	Piriformis
4	F	61	Post-traumatic		5.0	Femur	Piriformis
5	M	63	Post-traumatic		3.0	Femur	Not reported
6	M	65	Post-traumatic	Obesity	4.0	Femur	Retrograde
7	F	66	Post-traumatic	Remote infection, depression	3.0	Tibia	Antegrade
8	M	67	Post-traumatic		3.6	Tibia	Antegrade
9	M	69	Prior knee arthrodesis	Peripheral neuropathy, CAD	1.5	Femur	Retrograde
10	M	71	Post-traumatic		3.0	Femur	Retrograde
11	F	72	Post-traumatic	Obesity, prior infection	2.5	Femur	Trochanteric
12	M	67	Congenital		3.6	Tibia	Antegrade
13	F	61	Post-traumatic		2.5	Femur	Not Reported
14	F	63	Nonunion	Infection	5	Tibia	Antegrade
15	M	63	Genu-varum		1.5	Tibia	Antegrade
16	F	69	Post-traumatic		2.5	Femur	Piriformis

CAD, coronary artery disease; LLD, limb length discrepancy.

**Table 2 jcm-11-05242-t002:** Healing parameters.

	Consolidation Days		Consolidation Index		Maturation Days		Maturation Index		Distraction Index	
	Mean	95% CI	Mean	95% CI	Mean	95% CI	Mean	95% CI	Mean	95% CI
All Subjects (*n* = 354)	141	134, 148	34	32, 36	69	63, 75	17	16, 19	0.63	0.54, 0.72
Age										
7–19 (*n* = 261)	143	136, 150	33	31, 36	69	63, 75	17	15, 19	0.62	0.54, 0.72
20–39 (*n* = 55)	136	114, 158	36	30, 41	68	47, 89	19	14, 25	0.64	0.52, 0.71
40–59 (*n* = 22)	136	105, 166	41	29, 52	70	47, 93	21	13, 30	0.62	0.56, 0.73
60+ (*n* = 16)	141	106, 175	34	28, 40	69	49, 88	17	12, 22	0.65	0.54, 0.71
Sex										
Male (*n* = 186)	141	132, 150	34	30, 38	69	61, 77	18	15, 21		
Female (*n* = 168)	142	132, 153	35	32, 37	69	61, 78	17	15, 19		
Bone										
Femur (*n* = 259)	134	126, 142	31	29, 33	64	58, 71	15	14, 17	0.69	
Tibia (*n* = 95)	163	151, 175	43	37, 49	82	72, 92	23	18, 28	0.51	

CI, confidence interval.

**Table 3 jcm-11-05242-t003:** Lengthening outcomes of patients aged ≥ 60 years.

Patient	Distraction Index	Length Achieved (mm)	At Goal	Explanted (Yes/No)	Complications Post-Explanation
1	0.47	60	Yes	Yes	Infection
2	0.51	44	5 mm under	No	n/a **
3	0.77	79	2.3 cm over *	Yes	None reported
4	0.89	50	Yes	Yes	None reported
5	0.81	30	Yes	Yes	None reported
6	0.73	40	Yes	No	Patient deceased
7	0.79	30	Yes	No	n/a
8	0.29	36	Yes	Yes	Delayed union
9	0.65	15	Yes	No	n/a
10	0.65	30	Yes	Yes	None reported
11	0.57	25	Yes	Yes	None reported
12	0.56	32	Yes	Yes	None reported
13	0.70	20	No	No	n/a
14	0.65	25	No	No	n/a
15	0.72	15	Yes	Yes	Periosteal reaction
16	0.80	25	No	No	n/a

* Patient deliberately overlengthened. ** Not applicable since nail was not explanted.

**Table 4 jcm-11-05242-t004:** Complications.

	Older ≥ 60 (16)	Younger 7–19 (261)	Younger 20–39 (55)	Younger 40–59 (22)
Infection	1	6	0	0
Wound healing	0	5	1	2
Compartment syndrome	0	1	1	0
Fracture	0	4	1	0
Axial deviation	0	4	0	0
Lost length	0	1	1	2
DVT *	0	1	0	0
Subluxation	1	10	1	0
Nail fracture	1	6	2	0
Failure to lengthen	0	1	0	0
Symptomatic hardware	0	16	3	3
Rod fracture/failure	2	3	3	0
Delayed union	1	10	7	5
Malunion or nonunion	2	3	1	0
Contracture	0	45	18	3
Pre-consolidation	1	3	1	3
Not at goal length	3	3	1	0
Screw failure/revision	0	5	0	1
Acute distraction	0	1	0	0
Nerve	2	10	3	1
Gait deviation	0	1	1	0
Patients with complications	9/16 (56.3%)	139/261 (53.3%)	29/55 (52.7%)	13/22 (59.1%)
Patients with more than one complication	3/9 (33.3%)	42/139 (30.2%)	9/29 (31.0%)	4/13 (30.8)

* DVT, deep vein thrombosis.

## Data Availability

The data presented in this study are available upon request from the corresponding author. The data are not publicly available due to privacy concerns related to protected health information.

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
