# Peer review of "Motorized Intramedullary Nail Lengthening in the Older Population"

_jcm, 2022, doi:10.3390/jcm11175242_

Round 1

Reviewer 1 Report

Well done, well written paper on an important topic. I believe this study provides a significant contribution to the literature showing that limb lengthening can be considered in appropriately selected older patients. Despite some limitations in power because of numbers, it is a multicenter study from experts in the field and will be the best available evidence for this procedure in this population. 

I do think a few clarifications/changes may be beneficial to the strength of the study. 

1. Although all patients in the time frame were included, I think it would be helpful to define in the text what you would consider "clinically similar" (line 28) complication rates and see if the differences in complication percentage were statistically similar (i.e. what is the p value of the difference in 64% complication rate v 55%, line 166-169). 

I would also consider a post-hoc power analysis to determine what percent difference in complication rates you would have been powered to find (i.e. if you find these values are statistically similar (P>0.05), how big a difference would you have been powered to pick up with a study of this size) -- hard to definitively conclude they are the same unless you were powered to detect a 9% difference in complication rate, or don't feel that is a clinically significant difference based on how you define it. 

2. I was a little lost with the statistical 2:1 grouping (line 108) -- most of the tables seem to use all the patient data so I'm wondering for which analyses these groups were used for, why the 2:1 ratio was chosen, and if you can elaborate on what variables were propensity matched/present the demographic data that shows these were otherwise similar groups excepting the main variable, age. I would make this info a little more clear in the manuscript. 

3. I would report the categorical state in Table 4 as percentages. I think the decision not to run stats on each individual complication is okay given the small numbers, but it's difficult to even qualitatively compare number of events between groups of different sizes without a ratio. Again, I'd run the stats on the overall percent complications in different groups. 

4. In table 4 it says there was a 71% rate of complications in segments in the older group, but line 166 says 64%. I'm not sure if this is an error or they're referring to different things, but I would clarify. 

5. If you have the data available in the younger cohort, I'd look at differences in reoperation rate as a more objective obstacle end point (i.e is the percent of complications needing reoperation as opposed to just obs/PT/abx higher in the older group). I mention this because Table 3 seems to indicate that 6/16 patients required reoperation--I would comment on if it is a higher than expected amount compared to the younger group or based on your experience if it is about the same, this since as a surgeon doing this procedure I would find that number a bit concerning, but the conclusion may be that they can still be quite successful we just need to counsel patients that further intervention may be needed. 

6. Mean DI is reported for the older group but not for the younger (line 153), and that value is not included in Table 2 as it is for MI and CI. I would include this as well as a comment on if mean DI was or was not different between groups. 

Author Response

Well done, well written paper on an important topic. I believe this study provides a significant contribution to the literature showing that limb lengthening can be considered in appropriately selected older patients. Despite some limitations in power because of numbers, it is a multicenter study from experts in the field and will be the best available evidence for this procedure in this population. 

I do think a few clarifications/changes may be beneficial to the strength of the study. 

  1. Although all patients in the time frame were included, I think it would be helpful to define in the text what you would consider "clinically similar" (line 28) complication rates and see if the differences in complication percentage were statistically similar (i.e. what is the p value of the difference in 64% complication rate v 55%, line 166-169). 

Dear reviewer, thank you for your valuable comment. We have replaced wording “clinically similar” with a p-value for clarification. (line 30)

I would also consider a post-hoc power analysis to determine what percent difference in complication rates you would have been powered to find (i.e. if you find these values are statistically similar (P>0.05), how big a difference would you have been powered to pick up with a study of this size) -- hard to definitively conclude they are the same unless you were powered to detect a 9% difference in complication rate, or don't feel that is a clinically significant difference based on how you define it. 

Dear reviewer, thank you for your valuable comment. We did indeed run a statistical analysis on the age groups presented in Table 4. The percentage in complication change was not statistically significant (as originally proposed), hence no power analysis was necessary to determine the % difference. The data and explanation can be found in Table 4 (updated with percentages) and lines 197-200.  

  1. I was a little lost with the statistical 2:1 grouping (line 108) -- most of the tables seem to use all the patient data so I'm wondering for which analyses these groups were used for, why the 2:1 ratio was chosen, and if you can elaborate on what variables were propensity matched/present the demographic data that shows these were otherwise similar groups excepting the main variable, age. I would make this info a little more clear in the manuscript. 

The propensity matching was performed to identify comparable cohorts among the younger population who undergone limb lengthening. The model parameters were amount of lengthening, bone lengthened, size of the nail and gender. These editions were made in the methods section lines 113-115.

  1. I would report the categorical state in Table 4 as percentages. I think the decision not to run stats on each individual complication is okay given the small numbers, but it's difficult to even qualitatively compare number of events between groups of different sizes without a ratio. Again, I'd run the stats on the overall percent complications in different groups. 

 Dear reviewer, thank you for your valuable comment. We performed a chi-square analysis on the overall complications in different age groups and reported it in Table 4, as well as the results section lines 179-186.

  1. In table 4 it says there was a 71% rate of complications in segments in the older group, but line 166 says 64%. I'm not sure if this is an error or they're referring to different things, but I would clarify. 

Dear reviewer, thank you for your valuable comments. Some patients were added and others excluded throughout the process of writing this manuscript. The accurate account used in the study is now reflected in the manuscript as well as the tables. (Table 4)

  1. If you have the data available in the younger cohort, I'd look at differences in reoperation rate as a more objective obstacle end point (i.e is the percent of complications needing reoperation as opposed to just obs/PT/abx higher in the older group). I mention this because Table 3 seems to indicate that 6/16 patients required reoperation--I would comment on if it is a higher than expected amount compared to the younger group or based on your experience if it is about the same, this since as a surgeon doing this procedure I would find that number a bit concerning, but the conclusion may be that they can still be quite successful we just need to counsel patients that further intervention may be needed. 

Dear reviewer, thank you for your valuable comments. This is a great suggestion. From our experience, patients undergoing limb lengthening most frequently require re-operation either due to malunion/nonunion, mechanical nail failure or infection. This will be a great data set for further publications. As you pointed out, the patients had a high rate of successful surgery with only one requiring patient not reaching lengthening goal. In this manuscript, we replaced re-operation rates with nail explantation data and post-explantation complications to better assess the success of lengthening.  These are reported in Table 3.

  1. Mean DI is reported for the older group but not for the younger (line 153), and that value is not included in Table 2 as it is for MI and CI. I would include this as well as a comment on if mean DI was or was not different between groups. 

Dear reviewer, thank you for your valuable comments. We have included the values for a younger cohort which was not different from the rest of the age groups. This was also reflected in the results. (line 159-162)

Reviewer 2 Report

Dear authors,

Thank you very much for the possibility of reviewing this interesting manuscript which is the first study evaluating intramedullary lengthening for correction of leg length discrepancies in the geriatric population. The content of the paper is relevant to the readership and falls within the scope of the journal. The manuscript is well-written and concise, and I much enjoyed reading it.

I only have few remarks to make:

There is no mention whether nails were implanted through an ante- or retrograde approach. I assume all tibial nails were antegrade. But were all femoral nails retrograde, as in the radiographs provided, or did you only choose a retrograde approach if preceding surgeries precluded application of antegrade femoral nails? Maybe you could explain which approaches were employed in which cases. This would be interesting to know as the chosen approach determines the level of osteotomy, which in turn may affect the consolidation rate.

l. 130-133: This is a bit confusing to the reader. Patients were allowed full weight-bearing when 2/4 cortices were bridged, but still had to remain partially weight-bearing until 3/4 cortices were healed – this sounds somewhat contradicting? Maybe you could clarify.

Table 2:

- The total number of all age groups does not add up (261+55+22+10 would be 348, but the number of all subjects is 356).

- In line 135 you state that 188 MILNs were placed in male patients, but in Table 2 the number of male patients is 183? Also, the total of male and female patients does not add up (183 males plus 165 females would be 348, but the total number of MILNs was 356).

Table 3, “I&D”: Maybe you could define this abbreviation in the Table legend, as not every reader may be familiar with it?

Line 195: Redundant “lateral”.

Line 200: ROM of the knee joint, I assume?

Line 204: “regenerate.,” – redundant “.”.

Line 272: Do you mean six tibias? (See also line 136: “Seventeen MILNs (eleven femora, six tibiae”.)

Line 274-276: You state that additional research will be needed to assess the outcome after nail removal, which will be interesting. Maybe you could (in the Results section) give a short statement whether any nails had already been removed, and if so, if any adverse events were observed after removal?

Author Response

Dear authors,

Thank you very much for the possibility of reviewing this interesting manuscript which is the first study evaluating intramedullary lengthening for correction of leg length discrepancies in the geriatric population. The content of the paper is relevant to the readership and falls within the scope of the journal. The manuscript is well-written and concise, and I much enjoyed reading it.

I only have few remarks to make:

There is no mention whether nails were implanted through an ante- or retrograde approach. I assume all tibial nails were antegrade. But were all femoral nails retrograde, as in the radiographs provided, or did you only choose a retrograde approach if preceding surgeries precluded application of antegrade femoral nails? Maybe you could explain which approaches were employed in which cases. This would be interesting to know as the chosen approach determines the level of osteotomy, which in turn may affect the consolidation rate.

Dear reviewer, thank you for your valuable comments. We included information on nail insertion technique in the results section as well as in Table 1. As you have pointed out, retrograde nails were used in the instances where a different method would not be feasible due to previous surgery or existing hardware positioning. (Line 148-150)

  1. 130-133: This is a bit confusing to the reader. Patients were allowed full weight-bearing when 2/4 cortices were bridged, but still had to remain partially weight-bearing until 3/4 cortices were healed – this sounds somewhat contradicting? Maybe you could clarify.

Dear reviewer, thank you for your valuable comments. To clarify, patients are allowed full weightbearing when ¾ cortices are healed, as determined by the radiographs on follow-up. The previous sentence talking about bridged cortices was referring to partial weightbearing instructions. This sentence has been restructured in the manuscript on lines 134-136.

Table 2:

- The total number of all age groups does not add up (261+55+22+10 would be 348, but the number of all subjects is 356).

Dear reviewer, thank you for your valuable comments. Some patients were added and others excluded throughout the process of writing this manuscript. The accurate account used in the study is now reflected in the manuscript as well as the tables. (Table 2)

- In line 135 you state that 188 MILNs were placed in male patients, but in Table 2 the number of male patients is 183? Also, the total of male and female patients does not add up (183 males plus 165 females would be 348, but the total number of MILNs was 356).

Dear reviewer, thank you for your valuable comments. Some of the data was updated in the process of writing the manuscript, as in the previous comment. Please see Table 2 for edited information.

Table 3, “I&D”: Maybe you could define this abbreviation in the Table legend, as not every reader may be familiar with it?

Dear reviewer, thank you for this suggestion. We have now defined this abbreviation. (Line 218)

Line 195: Redundant “lateral”.

Dear reviewer, thank you for this comment. Redundant ‘lateral’ has been removed

Line 200: ROM of the knee joint, I assume?

Dear reviewer, thank you for this comment. We have clarified this sentence is referring to knee joint ROM in line 318.  

Line 204: “regenerate.,” – redundant “.”.

Dear reviewer, thank you for this comment. The redundant word has been removed.

Line 272: Do you mean six tibias? (See also line 136: “Seventeen MILNs (eleven femora, six tibiae”.)

Dear reviewer, thank you for this comment. Indeed, these numbers have changed since writing of the manuscript as per your previous comment. We have now updated the manuscript to reflect this.

Line 274-276: You state that additional research will be needed to assess the outcome after nail removal, which will be interesting. Maybe you could (in the Results section) give a short statement whether any nails had already been removed, and if so, if any adverse events were observed after removal?

Dear reviewer, this is an excellent suggestion. We have included information about explantation and complications post-explantation in Table 3 as well as in the results section.

Reviewer 3 Report

the paper is presenting 17 patients older than 60 treated by IMN lengthening using Precice motorized nail 

This device is now under prohibition by health authorities and FDA due to several complications 

The title is talking about Great ! , I do not think 3-6 cm is a great lengthening and this is not appropriate for a scientific paper title 

Author Response

the paper is presenting 17 patients older than 60 treated by IMN lengthening using Precice motorized nail 

This device is now under prohibition by health authorities and FDA due to several complications 

The title is talking about Great ! , I do not think 3-6 cm is a great lengthening and this is not appropriate for a scientific paper title 

Dear reviewer, you are correct that one of the lengthening nails has been taken off the market, however you are referring to the Precice Stryde stainless steel nail rather than the Precice titanium lengthening nail. These are completely different nails. The company originally developed the stainless-steel nail due to its ability to withstand earlier weightbearing, however, it has been associated with osteolysis and increased periosteal reaction and since has been taken off market. Patients with stainless steel lengthening devices were excluded from this study.

The title of the paper is referring to making greater effort to assess elderly patients for lengthening possibility rather then dismissing the case all together. It is a play on words and does not explicitly state that the lengthening is great.

Round 2

Reviewer 3 Report

I still recommend rejection due to the major issues about the technique and instruments used in the research 

Author Response

Dear Reviewer 3, we are sorry you remain disinclined to publish our paper. We feel that our work is original, and has potential to impact on the clinical practices of practicing orthopedic surgeons. There is a bias against the elderly for performing what some may call elective reconstructive surgery. “Why lengthen the leg, when the patient can simply use a shoe lift?”  Our younger population is generally unwilling to accept shoe lifts as a life long solution, for both cosmetic and functional reasons. Why should our elderly (>60 years old), who are living longer and longer these days, be forced to endure a Limb Length Discrepancy, because of their age.